# Pediatric Autoimmune Neuropsychiatric Disorders Associated with Streptococcal Infections (PANDAS) Syndrome: A 10-Year Retrospective Cohort Study in an Italian Centre of Pediatric Rheumatology

**DOI:** 10.3390/microorganisms12010008

**Published:** 2023-12-19

**Authors:** Saverio La Bella, Marina Attanasi, Armando Di Ludovico, Giovanna Scorrano, Francesca Mainieri, Francesca Ciarelli, Federico Lauriola, Luisa Silvestrini, Virginia Girlando, Francesco Chiarelli, Luciana Breda

**Affiliations:** Department of Pediatrics, “G. D’Annunzio” University of Chieti-Pescara, 66100 Chieti, Italy

**Keywords:** PANDAS, PANDAS syndrome, streptococcus, pediatric autoimmune neuropsychiatric disorders associated with streptococcal infections, antibiotics, GAS

## Abstract

Background. Pediatric Autoimmune Neuropsychiatric Disorders Associated with Streptococcal Infections (PANDAS) syndrome is a rare pediatric disorder consisting of a sudden onset of obsessive–compulsive disorder (OCD) and/or tics after a group A Streptococcus (GAS) infection. Methods. In the period between 2013 and 2023, 61 children presented to our Pediatric Rheumatology unit with a suspicion of PANDAS syndrome. Among these, a retrospective analysis was conducted, and 19 fulfilled the current classification criteria and were included in this study. Results. The male-to-female ratio was 14:5, the median age at onset was 7.0 (2.0–9.5) years, and the median age at diagnosis was 8.0 (3.0–10.4) years. The median follow-up period was 16.0 (6.0–72.0) months. Family and personal history were relevant in 7/19 and 6/19 patients. Tics were present in all patients. Details for motor tics were retrospectively available in 18/19 patients, with the eyes (11/18) and neck/head (10/18) being most often involved. Vocal tics were documented in 8/19, behavioral changes in 10/19, and OCD in 2/19. Regarding the therapeutic response, all patients responded to amoxicillin, 12/13 to benzathine benzylpenicillin, and 7/9 to azithromycin. Conclusions. Our findings partially overlap with previous reports. Larger prospective studies are needed to improve treatment strategies and classification criteria.

## 1. Introduction

The characterization of Pediatric Autoimmune Neuropsychiatric Disorders Associated with Streptococcal Infections (PANDAS) is a highly contentious topic. PANDAS syndrome is defined as a sudden onset of obsessive–compulsive disorder (OCD) and/or tic disorder symptoms in prepubescent children because of a Group A beta-hemolytic Streptococcus (GAS) infection. GAS is a Gram-positive coccus that is frequently responsible for tonsillopharyngitis and skin infections in children, with substantial societal and economic costs [1,2]. It is estimated to cause about 15–30 percent of pharyngitis in patients aged 5–15 years, mostly during the winter [2,3]. GAS infections often manifest with a high-grade fever, an intense sore throat, headache, nausea, and vomiting [1,2]. GAS infections are commonly treated with beta-lactam antibiotics due to the paucity of antibiotic resistance. Suppurative complications of GAS infections include abscesses, sepsis, and necrotizing fasciitis. GAS is also feared for its nonsuppurative complications, including acute rheumatic fever, post-streptococcal glomerulonephritis, and PANDAS syndrome [4,5,6,7].

The precise incidence and prevalence are unknown, but it is widely accepted that PANDAS syndrome is a very rare disease, probably underestimated. Although, in recent years, there has been an increasing understanding of its pathophysiology, it remains still not completely clear [5,8,9,10,11,12,13,14]. In fact, there is a dearth of comprehensive data regarding its management and clinical characterization [15,16,17,18]. There exist numerous analogies between PANDAS and Sydenham’s chorea (SC), including molecular mimicry that has been proposed as a key factor in the development of both conditions, resulting in the spreading of various types of autoantibodies that target neuronal cells [13,14,19,20,21]. Furthermore, growing knowledge is arising due to a major understanding of the roles of Ca^2+^/calmodulin-dependent protein kinase II (CAMKII) and striatal cholinergic interneurons, resulting in disrupted dopamine regulation and increased dopamine release in the basal ganglia [5,11,13,14,19,20,22].

The current working PANDAS classification criteria were published in 1998 and have been widely discussed in an expert opinion in 2017, but they still lack adequate accuracy [4,6,15,23]. 

Establishing a definitive causative relationship between GAS infection and the onset or recurrence of neuropsychiatric symptoms in PANDAS syndrome can be challenging. The occurrence of both GAS infections and tics/OCD in children is common and they may coexist without a causative pathogenic correlation.

The working classification criteria for PANDAS syndrome have been summarized in Table 1.

Recently, it has been suggested that more specific criteria should be met to improve the rate of diagnosis [6,15,23,24]. These include the complexity of symptoms, the inability to include tics and/or OCD in other defined disorders, the adoption of a specific age range, and the presence of sudden behavioral changes. Such recommendations stressed once again the importance of a clear association with a GAS infection, detectable with a positive throat swab (rapid or cultural test), or the elevation of anti-streptolysin O (ASO) and/or anti DNAse-B titers (ADB) [5,6,15,24]. 

The lack of precise treatment recommendations can be attributed to the limited number of prospective randomized trials available [5]. As a result, current therapy strategies primarily rely on a small amount of limited studies and expert opinions [15,16,17]. It is widely accepted that PANDAS syndrome can consistently improve with non-pharmacological approaches [15]. However, the main pharmacological treatment strategies are based on antibiotic drugs, such as beta-lactam antibiotics and macrolides, that target GAS infection and probably reduce antibody activity with an unknown mechanism of action [25,26,27,28]. In addition, immunomodulatory drugs, including intravenous immunoglobulins (IVIgs), glucocorticoids, and others, have been reported to be effective in selected patients [17]. Plasma exchange may be used as rescue therapy, with the rationale of removing circulating autoantibodies [5,29].

As introduced above, treatment strategies are currently based on some expert opinions and small observational trials, while large trials have provided controversial results. Moreover, there is no agreement on the appropriate dosage of drugs used for PANDAS syndrome [5,18].

The demographic characteristics, clinical manifestation, therapeutic response, and outcome of children diagnosed with PANDAS syndrome in our Pediatric Rheumatology unit within the previous decade were examined in this retrospective cohort study.

## 2. Methods

We carried out a ten-year retrospective assessment of the patients with a suspicion of PANDAS syndrome who attended the Pediatric Rheumatology unit of the Department of Pediatrics of the “G. D’Annunzio” University—“SS. Annunziata” Public Hospital of Chieti, Italy (Abruzzo regional Centre of Pediatric Rheumatology) throughout a period spanning from 2013 to 2023. The goals of this study were the following:To ascertain how many children attending our Pediatric Rheumatology unit with a suspicion of PANDAS syndrome fulfilled the existing classification criteria for the disease.To characterize the epidemiological characteristics and clinical features of children diagnosed with PANDAS syndrome in our unit.To evaluate the therapeutic approaches employed in our unit for PANDAS patients over the past decade and to determine their rate of efficacy.

In the last ten years, a total of 61 children attended our center with a clinical suspicion of PANDAS syndrome. These patients were referred to our attention from other doctors (general doctors, pediatricians, pediatric neurologists, and neuropsychiatrists). 

The inclusion criteria overlapped with the current classification criteria for PANDAS syndrome published by Swedo et al. in 1998 [4]. The exclusion criteria were the absence of a clear correlation between GAS infection and the onset or recurrence of symptoms, an age at onset after the prepubertal age (8 years in females, 9 years in males), a diagnosis of a different neurological syndrome, and the absence of a suggestive phenotype (epilepsy, neurodevelopmental delay etc.). 

Additionally, to ensure a definitive diagnosis and evaluation of therapy efficacy, we exclusively included children who had undergone a minimum of six months of follow-up. Based on the inclusion criteria, 5 participants were excluded because they were not of prepubertal age. Additionally, 8 children were excluded as their clinical presentation did not align with PANDAS syndrome (children with epilepsy or neuropsychiatric conditions lacking tics or OCD). A total of 9 patients were excluded because there was no evidence of a concomitant or recent GAS infection. Lastly, 20 participants were lost to follow-up before a definitive diagnosis of PANDAS could be established. Hence, a total of 19 children were clinically diagnosed with PANDAS syndrome and were deemed eligible for participation in this study (Figure 1).

To better characterize the cohort of children with PANDAS syndrome, epidemiological data, clinical characteristics, blood tests, and antibiotic treatment response were retrospectively evaluated. Continuous data are expressed as median (interval range), and categorical data are presented as percentage and count. Statistical analyses were performed using SPSS version 25.0 for Windows software (IBM Corp, Armonk, NY, USA).

A written consent was obtained by the participants and their parents. This paper is exempt from ethical committee approval since (i) it was confined to anonymized and unidentifiable data routinely collected at the Pediatric Rheumatology unit of the “SS Annunziata” public hospital—“G. D’Annunzio” University of Chieti; (ii) the data analyzed for this study were retrospectively analyzed; and (iii) this study’s findings did not affect patient care.

## 3. Results

### 3.1. Demographic Data and Clinical Features

The male-to-female ratio was 14:5 (73.7–26.3%), the median age at disease onset was 7.0 (2.0–9.5) years, and the median age at diagnosis was 8.0 (3.0–10.4) years. The median follow-up time was 16.0 (6.0–72.0) months. A positive family history for rheumatologic and/or neuropsychiatric disorders was present in 7/19 patients, including vocal tics, autoimmune thyroiditis, myasthenia gravis, psoriasis, rheumatoid arthritis, vitiligo, and systemic lupus erythematosus. Patient personal history was found positive for rheumatologic and/or neuropsychiatric disorders in 6/19 patients, including febrile seizures, attention deficit hyperactivity disorder (ADHD), ataxia, autism spectrum disorder, juvenile idiopathic arthritis, and post-streptococcal glomerulonephritis. ASO and ADB titers were available in 18 and 10 patients, respectively. A total of 16/18 patients had elevated ASO titer, 9/10 had elevated ADB titer, and 8/10 had both (normal values: below 150 U/L). All manifested motor tics, and details were available in 18/19 of them. Specifically, tics involved the eyes in 11/18, the head and/or neck in 10/18, the upper limbs in 7/18, the mouth in 4/18, the lower limbs in 3/18, and 3/18 had abdominal tics. Vocal tics were present in 8/19 patients. In contrast, OCD was reported only in 2/19 patients and consisted of an obsessive fear of germs and compulsive washing of hands in one patient and obsessive–compulsive walking movements in the second one. Behavioral abnormalities were reported in 10/19 patients, mostly consisting of aggression, agitation, nervousness, and isolation. Other neurological and neuropsychiatric symptoms were found in 11/19 patients, including dysgraphia, dysarthria, anxiety, pollakiuria, headaches, and enuresis. Brain magnetic resonance imaging was performed in 7/19 patients (one showed a slight supratentorial ventricular asymmetry, two documented sinusitis, and the others were negative). An electroencephalography was delivered in 9/19 patients (two had right parietal irritative abnormalities, one showed minor focal abnormalities prevalent on the left, and the others were normal). In 14/19 patients, an echocardiogram was performed with no evidence of valvular damage or insufficiency. The epidemiological and clinical characteristics of the cohort are summarized in Table 2.

### 3.2. Therapeutic Response 

Three distinct approaches were identified as being utilized for patients diagnosed with PANDAS syndrome over the last decade, due to the lack of definitive therapeutic recommendations. Amoxicillin was proposed as the first line of treatment in 14 patients, AZT in 1 patient, and benzathine benzylpenicillin in 4 patients (Table 3). 

Oral amoxicillin, oral azithromycin (AZT), and intramuscular benzathine benzylpenicillin were used according to the clinical response, patient and parents’ preferences, and medical experience. Amoxicillin (50 mg/kg/day for 7–10 days) was given in 15/19 patients. The therapeutic response was good, with a partial improvement in 12/15 and a complete response in 3/15 children (Figure 2A). AZT (10 mg/kg/day for 3–5 days) was administered in 9/19 patients, with a partial response in 5/9 patients, no response in 2/9 patients, and a complete response in 2/9 patients (Figure 2B). Lastly, benzathine benzylpenicillin (600,000 UI in patients with less than 27 kg, and 1,200,000 UI in patients with more than 27 kg, every 21–28 days) was used for 13/19 patients, with very good outcomes. No response was noted in 1/13 patient, while 2/13 patients showed a partial response and 10/13 had a complete remission. The median period of benzathine benzylpenicillin administration was 14 months (1.0–82.0).

## 4. Discussion

PANDAS syndrome is defined as the abrupt onset and/or recurrence of OCD and/or tic disorders following a GAS infection. There is a paucity of accurate data available on PANDAS syndrome, with some major points that still need to be clarified. The prevalence of both GAS infections and neuropsychiatric symptoms, such as tic disorders, is high in children. PANDAS pathogenesis has not been fully elucidated, but molecular mimicry has been proposed as a key factor due to the similarities with SC and the findings of some experimental studies [4,5,9,10,11,19]. Indeed, antibodies produced against GAS epitopes presumably cross-react with self-tissue, particularly with some proteins expressed in the neurons of the basal ganglia. Different models have been proposed with autoantibodies directed against lysoganglioside, tubulin, and dopamine 1 and 2 receptors (D1R, D2R) following a GAS infection [10,11,12,19]. Animal studies have documented a strong adaptive immunity response after GAS exposure in mice, with the spreading of autoantibodies against the cerebellum and the subsequent onset of movement disorders and behavioral changes [5,10,14]. The disruption of the brain–blood barrier has been related to the onset of symptoms in mice, with a consequent autoantibody spreading to cerebral and cerebellar structures [13,22]. Recent observations have emphasized the involvement of CAMKII in PANDAS pathogenesis, as these enzymes have shown heightened activity in murine models following exposure to autoantibodies derived from the blood of patients with PANDAS syndrome and SC [10,30]. Also, studies conducted on children with PANDAS documented elevated CAMKII activation compared to controls, resulting in an elevated release of dopamine [10,30,31]. Moreover, CAMKII activity was also found to be even higher in patients with SC, with a possible dose-dependent mechanism [5,10,31]. CAMKII are responsible for neuronal pathways and behavioral control, and their activity has been associated with a regulatory function in extracellular and intracellular calcium concentrations. Nevertheless, the precise pathogenic mechanism by which antibody-mediated CAMKII activation occurs in human neuronal cells after a GAS infection remains elusive; it is highly probable that it is associated with modifications in dopamine neurotransmission [10]. Regarding the autoantibodies, previous studies have discussed a role for D1R, while more recent research has highlighted the centrality of D2R in the basal ganglia and cerebral cortex [11]. Imaging studies conducted in PANDAS children confirmed the immune inflammation of the basal ganglia. Indeed, an increased size of the basal ganglia was observed in a group of PANDAS patients compared to controls, and the caudate and lentiform nuclei were found inflamed in PET scans, with improvement after treatment with IVIg [32,33]. A relevant role has also been attributed to the cholinergic interneurons (CINs) due to their peculiar affinity for autoantibodies. Within the striatum, they have the aim of regulating other neurons of the basal ganglia and D1 and D2 expression. A disruption of these pathways has been associated with the development of tics, compulsions, obsessions, behavioral changes, and hyperactivity [5].

A second point of debate is the accuracy of the current classification criteria for PANDAS syndrome [4,5]. As discussed above, symptom onset must occur before puberty, there must be a relapsing or remitting course, and a clear temporal link between a GAS infection and clinical manifestations must always be confirmed. Despite the recommendations provided in 2017, some points are still debatable. It is often difficult to prove a clear relationship between GAS infections and the onset of neuropsychiatric symptoms, mostly because of the high prevalence of these phenomena in children. Moreover, there is no agreement and still confusion about the PANDAS syndrome, and this can also result in misdiagnosis, as confirmed in our study. An infection from GAS can be demonstrated with a rapid or cultural swab test or with an increase in ASO and/or ADB titers. In the past, there have been several studies assessing the relationship between GAS and PANDAS symptoms, including a large case–control study on 75,000 children [34]. In contrast, a broad prospective cohort study with more than 800 children with sore throats failed to demonstrate a clear relationship between GAS and PANDAS symptoms [35]. An extensive study on more than a million children was conducted in 2017 in Denmark, proving an increased risk of developing OCD, tic disorders, and other mental issues after a streptococcal infection [36]. Also, other longitudinal studies reached similar conclusions [37]. Instead, additional studies did not provide a clear association between GAS infections and PANDAS onset [38,39,40]. 

Appropriately interpreting ASO and ADB titers could be difficult. In fact, these biomarkers exhibit a unique onset and duration. ASO usually increases one week after GAS infection, peaks after three to five weeks, and then decreases six to eight weeks later, remaining elevated for at least six months. Instead, ADB begins to rise two weeks after infection and declines three months later, with a peak between six and eight weeks [5,41].

Usually, tics and/or OCD develop two to three days after a GAS infection, which can affect the respiratory tract, skin, deep tissues, and genital areas [42].

The clinical history is characterized by periods of exacerbation and remission, ranging from weeks to months. The onset is abrupt by definition, and it is common to observe an admission to emergency departments due to the clinical phenotype.

Motor and verbal tics are frequently accompanied by hyperactivity, urinary symptoms (urgency, pollakiuria), eating disorders, anxiety, isolation, and aggression [5,43,44].

As briefly discussed in the introduction, treatment strategies are not universally defined. There is a scarcity of randomized studies in comparison to placebos and validated data concerning medications for PANDAS, which has led to a lack of comprehensive recommendations. Moreover, at present, there are no established protocols concerning the appropriate selection and administration of antibiotic strategies [5,18]. In clinical practice, the treatment strategy is predominantly determined by personal experience [5,18]. However, beta-lactam antibiotics (amoxicillin, benzathine benzylpenicillin, and cephalosporins), AZT, and clindamycin have been used with good outcomes [5,18,25,26,27,28,45]. The role of tonsillectomy is controversial [46]. 

This study aimed to assess the diagnostic rate of the patients who presented to our pediatric rheumatology unit with a suspicion of PANDAS syndrome, characterize the demographic and clinical features, and assess the therapeutic response.

Among the 61 children presenting with a suspicion of PANDAS syndrome, only 19 fulfilled the current classification criteria. The most important reason for exclusion from this study was the lack of follow-up before a diagnosis was made. In addition, exclusion criteria also comprised clinical features manifesting after pubertal onset, a better-categorized diagnosis in other neurological syndromes, and the presence of not-relevant clinical phenotypes (epilepsy, neurodevelopmental delay, etc.).

Our findings, in part, overlap with previous observations. We found a strong male prevalence (14/19), similar to what has been documented before [4]. Moreover, the median age at onset was 7 years of age, with a reported mean age at onset in the literature ranging from 6.3 to 7.4 years [4,47]. The median follow-up period was 16.0 (6.0–72.0) months. In patients with available data from the retrospective collection, ASO titer was elevated in 16/18, ADB in 9/10, and both in 8/10. A positive throat swab was documented in 7/13 patients. Family history was positive for rheumatologic and/or neuropsychiatric diseases in 7/19 patients, while personal history was significant for the same disorders in 6/19 of them. Such diseases comprised vocal tics, autoimmune thyroiditis, myasthenia gravis, psoriasis, rheumatoid arthritis, vitiligo, and systemic lupus erythematosus in first-degree relatives, and febrile seizures, ADHD, ataxia, autism spectrum disorder, juvenile idiopathic arthritis, and post-streptococcal glomerulonephritis in patients. All patients had motor tics; 18/19 patients had been reported in detail. In particular, they mostly affected the eyes (11/18), upper limbs (7/18), and head or neck (10/18). Vocal tics were documented in almost half of patients (8/19), while OCD was reported in 2/19, with compulsive movements and hand washing. Other neuropsychiatric symptoms were highly prevalent (11/19), as were behavioral changes (10/19). Brain MRI, echocardiography, and electroencephalography resulted in normal results in most patients. No valvular involvement was documented, due to the GAS infection, and neither cerebral abnormalities nor an increase in basal ganglia sizes was noted. The therapeutic response was good, especially in amoxicillin and benzathine-benzylpenicillin, but also in AZT, which was associated with good improvements. As discussed above, because of a lack of established recommendations, treatment lines were decided primarily based on personal experience.

## 5. Limitations

This study has a number of limitations. First, this was a retrospective cohort study with the limitations of the data from the study design. Data were collected from the patients‘ reports and laboratory investigations available in our hospital. This could result in a lack of systematic collection, with some data missing, as discussed in the text and tables. Some of the clinical and laboratory characteristics are lacking due to the structure of the study model. Moreover, no statistical comparative analysis was conducted on the cohort, due to the heterogeneity and paucity of the sample. Treatment strategies have resulted in heterogeneity, and it could not have been possible to define a clear criterion for the definition of the first, second, or third line of treatment, which was purely based on physician experience. We decided to establish a minimum 6-month follow-up period to include patients in this study and increase the possibility of defining therapeutic responses. In addition, a therapeutic response was defined as a clinical improvement noted by both the pediatric rheumatologist and the parents of the patients in the absence of objectively determined laboratory data or a validated clinical score. We only evaluated the antibiotic response, but in our cohort, there were no children under immunosuppressive therapy. In addition, non-pharmacological approaches were not evaluated. 

## 6. Conclusions

PANDAS syndrome is a rare disease affecting prepubertal children with a sudden onset of tic disorders and/or OCD after a documented GAS infection. Current classification criteria still lack good sensitivity and specificity. This often results in misdiagnosis and confusion. In our study, 61 patients presented to our Pediatric Rheumatology unit with a suspicion of PANDAS syndrome, but only 19 fulfilled the current classification criteria. Among these, all showed tic disorders, mostly affecting the eyes and head or neck, with a considerable presence of vocal tics. OCD was less documented. Amoxicillin, AZT, and benzathine-benzylpenicillin often resulted in good improvement in the clinical features. Larger, randomized, controlled prospective cohort studies are necessary to better clarify the characteristics of PANDAS patients and their therapeutic responses.

## Figures and Tables

**Figure 1 microorganisms-12-00008-f001:**
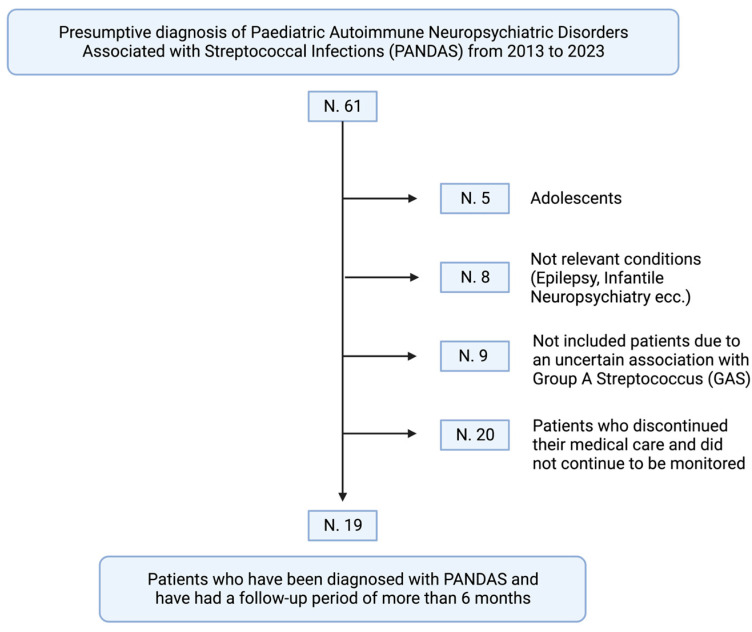
The retrospective process of inclusion of 61 children presenting to our center of Pediatric Rheumatology (created with BioRender.com). Based on the present classification criteria for PANDAS syndrome, the following patients were excluded: 5 had symptoms that appeared during or after puberty; 8 had clinical presentations that did not match the PANDAS syndrome (e.g., epilepsy, lack of tic disorder or OCD, complex patients with diagnosed neurological syndromes); 9 had an unconfirmed association with a Group A beta-hemolytic Streptococcus (GAS) infection; and 20 were lost to follow-up, resulting in the absence of a diagnosis. Thus, 19 children were ultimately diagnosed with PANDAS syndrome after a minimum of six months of follow-up.

**Figure 2 microorganisms-12-00008-f002:**
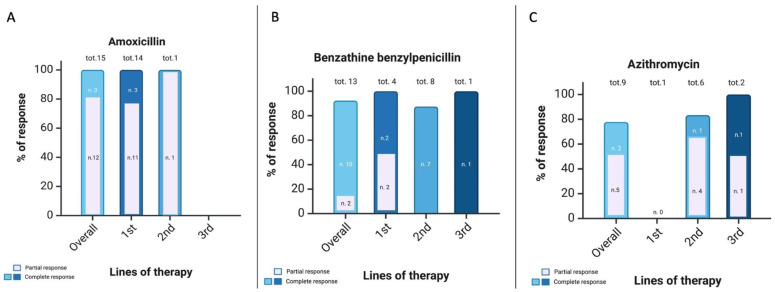
Treatment response to antibiotic drugs in the cohort of 19 patients diagnosed with PANDAS syndrome (created with BioRender.com). (**A**) Amoxicillin: 15 patients were treated with oral amoxicillin (50 mg/kg/day for 7–10 days), 14/15 as first-line treatment. All patients showed a good improvement; specifically, 12/15 had a partial response and 3/15 had a complete response. (**B**) Benzathine benzylpenicillin: 13 patients were treated with intramuscular benzathine benzylpenicillin (600,000 UI in children with less than 27 kg, 1,200,000 UI in children with more than 27 kg, every 21–28 days) for a median time of 14 months (1.0–82.0). A total of 4/15 were treated with benzathine benzylpenicillin as first-line treatments. A total of 12/13 showed a good improvement; specifically, 10/13 had a complete response and 2/13 had a partial response. A patient did not respond. (**C**) Azithromycin: 9 patients were treated with oral azithromycin (10 mg/kg/day for 3 days). A child was treated with azithromycin as first-line treatment with no response. A total of 7 patients showed a good improvement; specifically, 2/9 had a complete response and 5/9 had a partial response.

**Table 1 microorganisms-12-00008-t001:** Current classification criteria for PANDAS syndrome according to Swedo et al., 1998 [4]. Abbreviations. Obsessive–compulsive disorder: OCD; Group A beta-hemolytic Streptococcus: GAS; anti-streptolysin O: ASO; anti-DNAse B: ADB.

Working Classification Criteria for PANDAS Syndrome
1. Presence of OCD and/or tic disorders;
2. Symptoms must be related to a GAS infection through a positive throat swab or an elevation of ASO/ADB titers above the normal range;
3. Prepubertal symptom onset (mostly between 3 years and puberty);
4. Episodic course of symptom severity with abrupt onset/recurrence;
5. Association with neurological abnormalities (motor hyperactivity, choreiform movements, behavioral changes, etc.).

**Table 2 microorganisms-12-00008-t002:** Epidemiological and clinical characteristics of the 19 children diagnosed with PANDAS syndrome. Data are expressed as medians and range, and absolute numbers. Abbreviations. ASO: anti-streptolysin O; ADB: Anti-DNase B; OCD: obsessive–compulsive disorders; MRI: magnetic resonance imaging; EEG: electroencephalogram; * Family or Patient medical history positive for neuropsychiatric and/or rheumatologic disorders; ^§^ Dysgraphia, anxiety, pollakiuria, headache, enuresis, dysarthria; According to our laboratory data, normal values for ASO and ADB were considered below 150 U/L.

Variable	N.
Gender:	
- F (%)	5/19 (26.3)
- M (%)	14/19 (73.7)
Age at disease onset (years)	7.0 (2.0–9.5)
Age at diagnosis (years)	8.0 (3.0–10.4)
Follow-up period (months)	16.0 (6.0–72.0)
Troath swab, yes	7/13
Elevated ASO titer at diagnosis	16/18
Elevated ADB titer at diagnosis	9/10
Elevated ASO and ADB titers at diagnosis	8/10
* Family history, positive	7/19
* Patient medical history, positive	6/19
Tic Disorders, yes	19/19
- Motor tic, yes	18/18
- Ocular;	11/18
- Oral;	4/18
- Head and/or neck;	10/18
- Upper limbs;	7/18
- Lower limbs;	3/18
- Abdominal.	3/18
Vocal tic, yes	8/19
Behaviour modifications, yes	10/19
OCD, yes	2/19
^§^ Other neuropsychiatric symptoms, yes	11/19
Brain MRI, yes	7/19
EEG, yes	9/19
Echocardiogram, yes	14/19

**Table 3 microorganisms-12-00008-t003:** Treatment strategies and response in the cohort of 19 children diagnosed with PANDAS syndrome. Data are expressed as absolute numbers.

Antibiotic Drug	Lines of Treatment (N.)			
	First	Second	Third	Overall
Amoxicillin	14	1	0	15
- No response	0/14	0/1	-	0/15
- Partial response	11/14	1/1	-	12/15
- Complete response	3/14	0/1	-	3/15
Azithromycin	1	6	2	9
- No response	1/1	1/6	0/2	2/9
- Partial response	0/1	4/6	1/2	5/9
- Complete response	0/1	1/6	1/2	2/9
Benzathine benzylpenicillin	4	8	1	13
- No response	0/4	1/8	0/1	1/13
- Partial response	2/4	0/8	0/1	2/13
- Complete response	2/4	7/8	1/1	10/13

## Data Availability

The data presented in this study are available on request from the corresponding author.

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
