# Peer review of "Pediatric Autoimmune Neuropsychiatric Disorders Associated with Streptococcal Infections (PANDAS) Syndrome: A 10-Year Retrospective Cohort Study in an Italian Centre of Pediatric Rheumatology"

_microorganisms, 2023, doi:10.3390/microorganisms12010008_

Round 1
Reviewer 1 Report
Comments and Suggestions for Authors
Comments to microorganisms-2775931:
The manuscript is a retrospective analysis of the epidemiological characteristics and clinical features of Neuropsychiatric Disorders Associated with Streptococcal Infections (PANDAS) syndrome in prepubertal children. The manuscript is in general clearly written. Extensive English editing is required.
Comments:
· Line 43: ++ should be in superscript: Ca2+
· Since the authors use the abbreviation CAMKII in the discussion, it should be defined in line 43.
· The first part of the Introduction is in bold. Please correct.
· Please provide some more information about how Streptococcal infection affects Ca2+/calmodulin-dependent protein kinase II in striatal cholinergic interneurons.
· Line 45: An indent should be done for the new paragraph.
· Line 55: Please specify the symptoms.
· Line 75: Use another word than "contrast".
· Line 76: How can you say that antibiotics "target molecular mimicry with an unknown mechanism of action", when their action mechanisms are well-characterized? Please rephrase.
· Line 79: Is methotrexate also used? If so, please add.
· Line 80: How frequent is plasma exchange used in treating this syndrome?
· Line 81: Rephrase the sentence: " However, it is useful to repeat once again that"
· Line 106: Correct to "syndrome" (delete the plural "s").
· Lines 106-107: Define the age of prepubertal age. Why was it important to exclude children in the beginning of puberty?
· Line 117: ASO and ADB should be defined first time used.
· Figure 1 needs English editing.
· Can PANDAS develop in children after a one-time Streptococcal infection, where there is no increase in either ASO or ADB titer?
· The study was performed on pediatric patients that have visited the rheumatological department, meaning that they have had signs of a rheumatological disorder such as rheumatoid arthritis. Else, they wouldn't have come to this department. This needs to be stated in the text. It is likely that there have been more PANDAS patients coming to other departments of the hospital. Can these be tracked?
· Lines 152-153: The percentages in parentheses seem not to be correct: 16/19 = is 84.2% (not 88.9%); 9/19 is 47.3% (not 90%) and 8/19 is 42.1% (not 80%). The same for Table 1. The percentages are misleading due to few patients.
· Table 1: What do you mean by "at baseline", when they are positive? What was the criterion for the titer being defined as positive?
· Line 185: You can't say "has been proposed" when this was given to the children. Please rephrase.
· Table 2 is confusing since each child has got either one or more antibiotics. It should be divided into first-line, second-line and third-line. The response to first-line, second-line and third-line, etc. should be provided. The number of patients should be presented without the ".0".
· Due to few patients (N=19), presenting data in Figure 2-4 as percentages is misleading. It would be better to present the data in Table 2 where the sequential treatment is presented with the outcome.
· Can PANDAS syndrome develop after puberty?
· What is the percentage of children that have had GAS who develop PANDAS?
· What is the percentage of children that have ASO/ADB who develop PANDAS?
· Line 284: Correct to "6.3".
· I would remove all percentage numbers from the manuscript, due to the small patient number. Better to write the N/19.
· Did all the children respond at the end?
· How did the antibiotic treatment affect the ASO/ADB titers?
· If the antibiotics improved the conditions, it means that the children had a current undiagnosed infection. Please comment on it. Or the antibiotics had other effects besides combating infection?
Comments on the Quality of English Language
The manuscript needs English editing.
Author Response
The manuscript is a retrospective analysis of the epidemiological characteristics and clinical features of Neuropsychiatric Disorders Associated with Streptococcal Infections (PANDAS) syndrome in prepubertal children. The manuscript is in general clearly written. Extensive English editing is required.
Comments:
- Line 43: ++ should be in superscript: Ca2+
Done.
- Since the authors use the abbreviation CAMKII in the discussion, it should be defined in line 43.
Done.
- The first part of the Introduction is in bold. Please correct.
Done.
- Please provide some more information about how Streptococcal infection affects Ca2+/calmodulin-dependent protein kinase II in striatal cholinergic interneurons.
Thank you for your comment. It is known that in PANDAS syndrome and Sydenham's chorea autoantibodies target human neuronal cells and activated CAMKII. Murine models immunized with human monoclonal autoantibodies derived from patients with PANDAS and Sydenham's chorea showed an abnormal activation of CAMKII, leading to an increased dopamine release. Also studies conducted on children with PANDAS documented an elevated CAMKII activation compared to controls. Nevertheless, the precise pathogenic mechanism by which antibody-mediated CAMKII activation occurs in human neuronal cells subsequent to a GAS infection remains elusive; it is highly probable that it is associated with modifications in dopamine neurotransmission. We have added this point in Chapter 4.
- Line 45: An indent should be done for the new paragraph.
Done.
- Line 55: Please specify the symptoms.
Done.
- Line 75: Use another word than "contrast".
Done.
- Line 76: How can you say that antibiotics "target molecular mimicry with an unknown mechanism of action", when their action mechanisms are well-characterized? Please rephrase.7
Done.
- Line 79: Is methotrexate also used? If so, please add.
No, according to our experience and literature analysis there is no mention of methotrexate use in PANDAS children.
- Line 80: How frequent is plasma exchange used in treating this syndrome?
It is used as rescue therapy for non-respondent children. It has been specified in the text.
- Line 81: Rephrase the sentence: " However, it is useful to repeat once again that"
Done
- Line 106: Correct to "syndrome" (delete the plural "s").
Done.
- Lines 106-107: Define the age of prepubertal age. Why was it important to exclude children in the beginning of puberty?
Done, normally puberty onset occurs after 8 years in females and 9 years in males. Prepubertal onset is mentioned in the classification criteria by Swedo SE et al. (1998) and in an expert opinion published by Chiarello F et al. in 2017 .
- Line 117: ASO and ADB should be defined first time used.
Thank you for your comment. ASO and ADB have been defined in lines 66-67.
- Figure 1 needs English editing.
Thank you for your suggestion. Figure 1 has been modified.
- Can PANDAS develop in children after a one-time Streptococcal infection, where there is no increase in either ASO or ADB titer?
Thank you for your question. Yes, it is possible because GAS infection could also be detected through a throat swab; ASO and ABD rise in the majority of patients who suffered from a GAS infection, but the sole positive swab could be sufficient. Indeed, in our study, not all patients showed elevated ADB and/or ASO titers. However, positive ASO/ADB should decrease parallel to the improvement of symptoms.
- The study was performed on pediatric patients that have visited the rheumatological department, meaning that they have had signs of a rheumatological disorder such as rheumatoid arthritis. Else, they wouldn't have come to this department. This needs to be stated in the text. It is likely that there have been more PANDAS patients coming to other departments of the hospital. Can these be tracked?
Thank you for your comment. PANDAS syndrome is usually diagnosed and managed by pediatric neurologists and pediatric rheumatologists, similar to Sydenham chorea and rheumatic fever. Our pediatric rheumatology unit is part of the Department of Pediatrics, and patients coming to our attention could also have been referred to the pediatric neurologists or the general pediatrician of the hospital and then referred to us for diagnosis and treatment.
- Lines 152-153: The percentages in parentheses seem not to be correct: 16/19 = is 84.2% (not 88.9%); 9/19 is 47.3% (not 90%) and 8/19 is 42.1% (not 80%). The same for Table 1. The percentages are misleading due to few patients.
Thank your for your comment. There were some errors in the text; for example, percentages were correct but those values should have been “9/10” instead of "9/19" and “8/10” instead of "8/19" because we did not have the full data on ASO and ADB titers due to the retrospective nature of this study. Some patients did laboratory exams in external laboratories, and we did not have full access to the data. However, percentages have been deleted from both the text and tables for a better understandability, as suggested.
- Table 1: What do you mean by "at baseline", when they are positive? What was the criterion for the titer being defined as positive?
Thank you for your comment. “At baseline” was replaced with “at diagnosis”. ADB and ASO titers were considered elevated when above 150 U/L, as in our laboratory reports.
- Line 185: You can't say "has been proposed" when this was given to the children. Please rephrase.
Done
- Table 2 is confusing since each child has got either one or more antibiotics. It should be divided into first-line, second-line and third-line. The response to first-line, second-line and third-line, etc. should be provided. The number of patients should be presented without the ".0".
Thank you for your suggestion. We have modified Table 2 based on your indications.
- Due to few patients (N=19), presenting data in Figure 2-4 as percentages is misleading. It would be better to present the data in Table 2 where the sequential treatment is presented with the outcome.
Thank you for your comment. We have realized a unique Figure panel as suggested by the other reviewer and removed percentages from the Table and the text. Also, we have removed percentages from both Figure and description.
- Can PANDAS syndrome develop after puberty?
This is a debatable point. However, the current classification criteria provided by Swedo et al. in 1998 (PMID 9464208) and the expert opinion published by Chiarello et al. in 2017 (PMID 28498087) excluded this hypotesis for the diagnosis.
- What is the percentage of children that have had GAS who develop PANDAS? What is the percentage of children that have ASO/ADB who develop PANDAS?
Thank you for your questions. Currently, there is no precise data on this topic. However, in a longitudinal study, ten children developed PANDAS among 30000 throat cultures positive for GAS (PMID 15351749). The discussion on ASO/ADB and PANDAS development is even more complex with no accurate data, but, for example, a further prospective study examined 693 children monthly and collected throat cultures and data on behaviors. The study found a correlation between repeated GAS infection (and thus an increase in ASO/ADB titers) and choreiform movements and behaviors, as well as between GAS infection in the previous three months.
- Line 284: Correct to "6.3".
Done.
- I would remove all percentage numbers from the manuscript, due to the small patient number. Better to write the N/19.
Done. Percentages were left only for sex.
- Did all the children respond at the end?
Thank you for your question. Yes, all patients had at least a partial response to one of the three treatment strategies. However, we did not include children with less than 6 months of follow-up for having an accurate diagnosis, and it is probable that some of these patients were lost at follow-up for inadequate responses to treatment.
- How did the antibiotic treatment affect the ASO/ADB titers?
Thank you for your question. Generally, ASO/ADB titers decreased in tandem with the clinical improvement brought about by the antibiotic treatment. This phenomenon has not been reported upon within our cohort as a result of an incomplete dataset. Nonetheless, data are accessible to other academics who pose a reasonable inquiry. Also, we have not discussed ASO/ADB dynamics in detail because it has been recently discussed in another paper on the topic in your journal. (https://doi.org/10.3390/microorganisms11102549).
- If the antibiotics improved the conditions, it means that the children had a current undiagnosed infection. Please comment on it. Or the antibiotics had other effects besides combating infection?
Thank you for your question. Some of the cohort patients had a positive throat swab and thus could have directly responded to the antibiotic treatment. However, the majority of them had elevated ASO/ADB titers with no signs of a concomitant GAS infection. The exact mechanisms of action of antibiotic drugs in PANDAS patients are still unclear; indeed, there are no standardized therapies, and the main pathogenic basis arises from the similarities between patients with Sydenham chorea. A recent exper opinion (PANDAS: Pediatric autoimmune neuropsychiatric disorder associated with group A streptococci - UpToDate) recommends, for example, to use azithromycin because of the possibility that GAS is intracellular and do not recomment amoxicillin for its low intracellular penetration.
English editing has been performed.
Reviewer 2 Report
Comments and Suggestions for Authors
The manuscript by La Bella S. et al is a retrospective study that evaluates the demographic characteristics, clinical phenotype, pharmacologic response, and clinical outcome of 19 prepubertal children with confirmed pediatric autoimmune neuropsychiatric disorders associated with Streptococcal infections (PANDAS) syndrome, collected in a 10-years.
All sections of the manuscript appear well written, but I suggest some revisions that can improve the quality of findings and the presentation of data:
- The introduction needs a paragraph on Streptococcus group A infection, the clinical manifestations, the incidence rate.
- Introduction can be implemented with further references on PANDAS, and the criteria used for its classification.
- Bullet points can be replaced in the introduction with an organic text.
- A single figure with 3 panels can include Fig. 2, 3 and 4, to permit better comparisons.
- Figures 2, 4 and 4 must be cited in the results section instead on in the discussion.
Comments on the Quality of English LanguageModerate editing of English language is required.
Author Response
The manuscript by La Bella S. et al is a retrospective study that evaluates the demographic characteristics, clinical phenotype, pharmacologic response, and clinical outcome of 19 prepubertal children with confirmed pediatric autoimmune neuropsychiatric disorders associated with Streptococcal infections (PANDAS) syndrome, collected in a 10-years.
All sections of the manuscript appear well written, but I suggest some revisions that can improve the quality of findings and the presentation of data:
- The introduction needs a paragraph on Streptococcus group A infection, the clinical manifestations, the incidence rate.
Thank you for your suggestion. A brief introduction to GAS has been added in the introduction.
- Introduction can be implemented with further references on PANDAS, and the criteria used for its classification.
Thank you for your comment. We have added more references on the disease and a new Table summarizing in detail the current classification criteria for PANDAS syndrome with the related references.
- Bullet points can be replaced in the introduction with an organic text.
Done.
- A single figure with 3 panels can include Fig. 2, 3 and 4, to permit better comparisons; figures 2, 3 and 4 must be cited in the results section instead on in the discussion.
Thank you for your suggestion. A unique figure panel has been created and moved to the results section.
Round 2
Reviewer 1 Report
Comments and Suggestions for Authors
The authors have addressed all issues and the manuscript is acceptable for publication.
Minor comment:
2+ in line 52 should be in superscript.